# A Breakthrough in the Treatment of Necrobiosis Lipoidica? Update on Treatment, Etiopathogenesis, Diagnosis, and Clinical Presentation

**DOI:** 10.3390/ijms25063482

**Published:** 2024-03-20

**Authors:** Maciej Naumowicz, Stefan Modzelewski, Angelika Macko, Bartosz Łuniewski, Anna Baran, Iwona Flisiak

**Affiliations:** Department of Dermatology and Venereology, Medical University of Bialystok, 15-540 Bialystok, Poland; 38741@student.umb.edu.pl (M.N.); 38738@student.umb.edu.pl (S.M.); 39019@student.umb.edu.pl (A.M.); 38730@student.umb.edu.pl (B.Ł.); iwona.flisiak@umb.edu.pl (I.F.)

**Keywords:** necrobiosis lipoidica, etiopathogenesis, treatment, biologics, Janus kinase inhibitors

## Abstract

Necrobiosis lipoidica (NL) is a rare granulomatous disease of a not fully understood etiopathogenesis. Classically, NL is associated with insulin-dependent diabetes mellitus. The disease often fails to respond to conventional treatments and adversely affects patients’ quality of life. First-line medications are usually topical corticosteroids, but patients respond to them with varying degrees of success. Other options include tacrolimus, phototherapy, cyclosporine, fumaric acid esters, and biologics (adalimumab, etanercept, and infliximab). Our review aims to present new therapeutic approaches potentially effective in patients with refractory lesions, describe the presumed etiopathogenesis, and provide diagnostic guidance for clinicians. The review concludes that Janus kinase inhibitors and biologics such as ustekinumab and secukinumab can be used effectively in patients with recalcitrant NL. Another promising treatment option is tapinarof (an aryl hydrocarbon receptor agonist). However, studies on larger groups of patients are still needed to evaluate the effectiveness of different therapeutic options and to define consistent treatment regimens for NL. It is advisable to improve the awareness of physicians of various specialties regarding necrobiosis lipoidica as lesions diagnosed earlier usually have a better response to treatment.

## 1. Introduction

Necrobiosis lipoidica (NL) is a rare granulomatous disease that affects women more often than men. It most commonly occurs between the ages of 30 and 50 years, but cases of onset at younger ages have been reported [1,2,3,4]. The disease is often associated with diabetes, but it can also occur in patients with other diseases as well as in healthy individuals [5]. NL is manifested as well-demarcated yellow-brown plaques, most often localized on the pre-shin area of the lower extremities, with atrophy of the epidermis in the central part and with visible dilated vessels [6]. To date, no conclusive treatment regimens for NL have been established; this is due to the incompletely understood etiopathogenesis and the rarity of the disease, which makes it difficult to conduct studies on the use of different therapeutic options in larger groups of patients. This review compiles current data on the etiopathogenesis, clinical presentation, and diagnosis of NL and discusses recent therapeutic approaches that have been shown to have beneficial effects in the treatment of NL.

## 2. Etiopathogenesis

Necrobiosis lipoidica was first described in 1929 by Oppenheim [7] and was, at the time, associated exclusively with diabetes. In 1935, the first case of a patient with NL without diabetes was described [8]. The etiopathogenesis of necrobiosis lipoidica still remains unexplained. The disease can coexist with thyroid dysfunction, immune-mediated diseases such as sarcoidosis and rheumatoid arthritis or metabolic syndrome [9], but it is most commonly associated with insulin-dependent diabetes. Moreover, 11–87% of people with NL are diagnosed with diabetes, while only 0.3–1.2% with diabetes develop NL [5]. Due to the fact that the disease is rare and the data are from case series reports, the frequency of association with diabetes varies significantly. Jockenhofer et al., considering 262 patients with NL, indicated that insulin-dependent diabetes mellitus was present in 20.6% (52) of patients [10]. They also referred to the fact that older case reviews did not consider components of the metabolic syndrome (hypertension or obesity) [11,12,13] and that the ratio of women to men differed in selected studies. In addition, it seems impossible to compare historical studies with the most recent ones, given the changes in lifestyle that affect the frequency of development of diabetes risk factors [14]. Despite the mentioned data, the close association of NL with diabetes remains controversial [13].

The histological picture of NL coexisting with diabetes (in this case, necrobiosis lipoidica diabeticorum, NLD) includes palisade granulomas surrounding areas of necrobiosis, while in NL unrelated to diabetes, tuberculoid lesions are more typically observed [1]. This diversity underscores the complexity of the disorder, in which it is possible to have similar-looking lesions that may have originally been initiated by different factors. Of the theories explaining the pathogenesis of NL, most studies support the view of the importance of microangiopathy and ongoing local inflammation and collagen degeneration.

By immunohistology, an increased number of glucose transporter 1 (GLUT-1, glucose transporter 1) has been visualized on fibroblasts, which are responsible for the production of collagen, essential for proper wound healing [15]. Interestingly, studies have shown that fibroblasts in the lesion themselves produce type I collagen, typical of wounds versus type III—differentiated collagen, in appropriate proportions, but the amount of collagen itself was lower than in healthy skin, and electron microscopy showed a loss of cross-linking of collagen fibers and irregularities in the diameter of the fibers themselves [16]. In addition, a reduced amount of mRNA for procollagen in fibroblasts was shown, which may be a clue in the pathogenesis of the lesion, suggesting the inability of fibroblasts to function normally. An attempt to explain this condition may be the previously mentioned increased amount of GLUT-1, where typically their up-regulation is seen when cellular energy demands increase. The condition of insulin-dependent diabetes indicates that such cells may not be sufficiently saturated with energy. However, it would then be plausible that regulations of the primary disease should lead to improvements in the patient’s condition. Data from the literature, however, are insufficient to determine whether glycemic regulation is crucial, but there are reports indicating significant improvement with pancreas transplantation or a change in insulin dosing regimens [17]. Erfurt-Berge et al. observed, in a retrospective analysis of 52 patients with NL, that the vast majority had features of the metabolic syndrome or diabetes, while a small percentage of patients had other diseases, which may further indicate a more homogeneous pathogenesis of the disorder [18]. Another significance of increased GLUT-1 expression may be its involvement in the development of vascular occlusion, leading to glycoprotein adhesion to the receptor and constriction of the vascular lumen, which would lead to impaired perfusion and reduced oxygen metabolism [15]. Histologic studies indicate vascular involvement, but this affects about one-third of the lesions [13].

The results of flow and oxygenation measurement studies do not give a clear answer about possible tissue hypoxia in the course of NL [19,20]. There are data that suggest that the lesion is not worse but better oxygenated than healthy skin, which would be the result of the inflammatory process. In studies of sections from NL skin lesions, an infiltrate of granulomatous components surrounding the vasculature has been observed, including such cells as macrophages and lymphocytes [21]. There are also reports of the presence of IgM in the involved vessels, as well as C3 components within the dermal–epidermal junctions [22].

Since NL is often diagnosed with the coexistence of metabolic syndrome, as mentioned above, and diabetes, there is a need to take a closer look at insulin resistance (IR), which is crucial for those diseases [23,24]. Moreover, IR was found to be one of the key factors in other dermatological diseases, like acne vulgaris [25]. IR promotes the synthesis of TNF-α and other inflammatory cytokines and can cause chronic general inflammation. TNF-α is also found in higher concentrations in blood and tissues taken from chronic non-healing wounds [26]. Miele et al. noted that despite high glucose levels in people with type 2 diabetes, there were elevated levels of GLUT-1, while the higher glucose-dependent GLUT-4 had a 3-fold reduction in expression on the surface of fibroblasts [27]. It should be noted that one of the actions of TNF-α is precisely the inhibition of GLUT-4 expression. In addition, it is important to know that GLUT-1 is also dependent on the levels of various growth factors, which increase when insulin levels are high [28]. This could also explain the efficacy of TNF-α inhibitor drugs in NL [29].

Attention has also been given to the role of genetic factors in the development of necrobiosis lipoidica, but data are insufficient to determine the role of genetics in this disorder [9]. Examples of familial NL are present in the literature—in siblings without diabetes [30], in siblings with diabetes [31], and in monozygotic twins with newly diagnosed type 2 diabetes in both [32]. Tissue compatibility system (HLA) testing for NLD associated with type 1 diabetes showed no differences between diabetics with NLD and those with diabetes only, calling into question the validity of including genetics in etiopathogenesis [33]. However, it should be noted that this study was limited to type 1 diabetes only and did not address cases of NLD associated with type 2 diabetes or NL not associated with diabetes.

## 3. Clinical Features

The disease begins with red papules or nodules, usually without subjective symptoms. These transform into round or oval plaques, which may be accompanied by pruritus and paresthesias [34]. According to Patel et al., 84% of patients have more than one lesion, and 50% have more than four [34]. The lesions enlarge peripherally, leaving a yellowish-brown atrophy in the central area with dilated small vessels, usually surrounded by a raised erythematous border. Within the lesions, there is often decreased skin sensation due to a reduction in the number of nerve endings compared to the unaffected surrounding skin [6]. In one-third of patients, the lesions may develop into ulcerations, particularly as a result of trauma [1]. They can cause pain, secondary infections, and the development of squamous cell carcinoma or even amputation [35]. However, cases of spontaneous resolution of ulcerations have been reported, occurring in up to 17% of patients [1]. NL lesions are localized to the lower extremities in 85% of cases. They are usually bilateral on the pretibial surface [36]. However, they may also occur in other areas. Unusual cases of lesions on the face, scalp, trunk, penis, and upper extremities have been reported [1]. Unusual localizations have also been described in pediatric patients, with lesions observed in the interscapular region, on the abdomen, and on the extensor surface of the upper arm [3,4,37,38]. In NL, Köbner’s sign may occur, and lesions appear in surgical scars and after skin trauma [9,39]. Rare variants of NL are also seen in clinical practice. Perforating necrobiosis lipoidica is most common in women between 30 and 60 years of age. It manifests with the presence of keratotic caps in areas typical of classic NL. After the removal of the lesion, characteristic small pinpoint depressions remain. In most cases, this variant of NL is associated with non-complicated and non-insulin-dependent diabetes mellitus. On the other hand, due to its unusual localization, periorbital necrobiosis lipoidica must be distinguished from necrobiotic xanthogranuloma associated with paraproteinemia, which most commonly occurs on the face, especially on the lower and upper eyelids [9,40]. Cases of NL on other parts of the face have also been described [41,42]. Necrobiosis lipoidica can occur at the site of trauma, surgical scarring, as a result of Koebner’s phenomenon [43]. Cases of NL have been described at burn wounds, in appendectomy scars, after breast reduction surgery or phlebectomy [39,44,45,46]. Necrobiosis lipoidica very rarely affects the genital area, although cases of NL on the penis and scrotum have been described [47,48,49]. This form of the disease is often distinguished from granuloma annulare (GA). Tokura et al. distinguished four differences between NL and GA in the intimate area [47]. The lesions in NL are limited to the glans, and in GA, to the shaft of the penis. Ulceration and atrophic scarring are typical of NL, while GA presents with a few nodules without ulceration. NL begins in older patients. Histologically, NL tends to have larger foci of necrosis, while in GA, the lesions contain more mucin.

## 4. Diagnosis

The diagnosis of NL is usually based on the clinical picture. However, because of the possible atypical course and diagnostic challenges in the early stages of the disease, histopathologic examinations of a cutaneous lesion should be considered [9]. Lepe et al. describe the formation of interstitial and palisade granulomas. These lesions are organized in a layered manner and are admixed with patches of collagen degeneration [50]. Necrotic collagen is most commonly found in the lower two-thirds of the dermis, and regenerating collagen fibers may be found adjacent. The necrotic areas are surrounded by inflammatory cells, mainly non-epithelial histiocytes, but also Langerhans giant cells, lymphocytes and fibroblasts [6]. There is also a reduction in the number of nerve fibers in areas of inflammation. In NL, the swelling of endothelial cells and thickening of blood vessel walls can also be observed, but these changes are also characteristic of diabetic microangiopathy [50]. Occasionally, swelling may be the cause of vessel blockages [51]. If ulceration is present, histopathological examinations should be repeated to rule out squamous cell carcinoma [9]. In cases of diagnostic difficulties, particularly in the early stages of the disease, a dermoscopy can be helpful. The abnormalities observed upon dermatoscopic examinations were presented by Shrestha et al. [52] (Table 1).

Tree-shaped branching vessels are observed on a yellow, structureless area, often with white streaks [53,54,55]. In the initial stages of the disease, the dermoscopic image displays comma-shaped vessels on a pink background, orange-brown areas, and a thin network of vessels in the upper part. In more advanced lesions, a network of vessels on a pink background with homogeneous orange-yellow areas is observed. In the final stage, tree-shaped branching vessels are seen on a light brown background, along with whitish areas and a heterogeneous pigmented network [56].

## 5. Differential Diagnosis

Non-infectious granulomatous skin diseases, which include NL, are a broad group of conditions with many similarities [57,58]. The classic form of NL has a characteristic clinical picture, but the atypical course and early stage of the disease can pose diagnostic problems [51]. The differences in clinical and histological presentation and features seen upon dermatoscopic examinations of the various granulomatous diseases and BCC are summarized in Table 2.

## 6. Treatment Standards and New Therapeutic Options

Due to the rarity of necrobiosis lipoidica in the population, no uniform treatment guidelines have been established to date. Topical glucocorticosteroids (GCSs), which are the most commonly used in the treatment of NL, had a positive effect in 40% (14/35) of uses in a multicenter study conducted by Erfurt-Berge et al. [62]. This positive effect was characterized by no increase in the number and surface area of lesions, no new ulcerations, and a reduction in active inflammation. Topical GCS can cause skin atrophy, which is why it is not advisable to apply these preparations on atrophic lesions [63,64]. Other side effects of topical GCS include striae, rosacea, perioral dermatitis, acne, purpura, hirsutism, pigmentation alternations, delayed wound healing, and aggravation of cutaneous infections [64]. Prolonged use of topical corticosteroids, particularly on a large surface area, can exacerbate hyperglycemia, which complicates glycemic control in diabetic patients. For this reason, the systemic use of GCS in patients with NL and diabetes is controversial [64,65]. Calcineurin inhibitors (especially tacrolimus) are also frequently used in the topical treatment of NL [62]. In a study by Erfurt-Berge et al., tacrolimus was found to be more effective than topical GCS, with a positive effect observed in 61.5% (8/13) of uses [62]. Tacrolimus has an advantage over GCS as it does not cause skin atrophy and can be applied to areas with atrophic lesions and on the face [66,67,68]. Additionally, the literature indicates that tacrolimus is particularly effective in treating NL ulcers [69,70]. Other therapeutic options include phototherapy, fumaric acid esters, or dapsone [62,71,72]. Antimalarials (chloroquine, hydroxychloroquine), cyclosporine, doxycycline, and pentoxifylline have also been used in the treatment of NL [62,72,73]. Biological treatment is used when other therapeutic options are ineffective or there are contraindications to the use of other drugs. Attempts to use biologics have mainly involved TNF-α inhibitors (adalimumab, infliximab, and etanercept) [72,74]. The discovery of a key role for TNF- α in granuloma formation in mouse models provides a theoretical basis for explaining the efficacy of TNF-α inhibitors in granulomatous inflammatory diseases such as NL [74,75]. Although medications in this class have had a beneficial effect on many patients, there were cases reported that did not respond to treatment or the treatment had to be discontinued due to the loss of efficacy or adverse effects [76,77,78,79]. This has necessitated the search for new drugs effective in the treatment of necrobiosis lipoidica. In recent years, cases have been described of the successful use of biologics with a different molecular target—ustekinumab and secukinumab (Table 3)—as well as Janus Kinase inhibitors (JAKi) and the aryl hydrocarbon receptor (AhR) agonist—tapinarof.

### 6.1. Ustekinumab

Ustekinumab is a fully human monoclonal antibody that binds IL-12 and IL-23 via the p40 subunit, which prevents the IL-12 activation of Th1 lymphocytes and their release of IFN-γ. This results in the inhibition of the interferon gamma (IFN-γ)-dependent pathway of macrophage activation, which is involved in granuloma formation [82], whereas IL-23 binding results in a lack of the stimulation of Th17 lymphocytes, which can be found in material collected from NL lesions [83]. Only single clinical case reports regarding the use of Ustekinumab in the treatment of necrobiosis lipoidica were found [76,77,80,81] (Table 3).

In three out of four of these, clinical improvement was observed in the absence of side effects. Two patients responded to Ustekinumab despite a previous history of TNF-α inhibitors, which were discontinued due to lack of response to treatment, loss of efficacy, or significant side effects [76,77]. Meanwhile, McPhie et al. described a 71-year-old patient with recalcitrant NL lesions who showed no improvement after 21 months of Ustekinumab; only the use of JAKi resulted in a significant clinical improvement [81]. The results of Ustekinumab in the treatment of NL appear promising, but multicenter studies in larger groups of patients are needed to assess its actual efficacy.

### 6.2. Secukinumab

Secukinumab is a fully human monoclonal antibody that binds IL-17A, preventing it from binding to the receptor for IL-17, consequently inhibiting the action of this pro-inflammatory cytokine [84]. Wakusawa et al., in sections taken from NL lesions, revealed the presence of granulomas containing significant amounts of IL-17-releasing cells; therefore, this cytokine was associated with NL [85]. Gibson et al. described four clinical cases of patients with treatment-resistant NL who received subcutaneous secukinumab for 24 weeks [78]. All patients responded to treatment, but the clinical improvement varied between patients (from about 25% to about 90% improvement). The authors suggest that secukinumab may be a potential, safe therapeutic option for the treatment of NL [86].

### 6.3. JAK Inhibitors

In recent years, clinical case reports with attempts to treat NL with JAK inhibitors off-label have been published.

#### 6.3.1. Potential Mechanism of Action of JAK Inhibitors in NL

The mechanism of action of JAK inhibitors in the treatment of NL has not been fully elucidated. Some cytokine receptors (including IFN-γ, IL-2, IL-6, IL-12, and IL-23) have no intrinsic kinase activity and use the JAK-STAT pathway to activate signaling [87,88] (Figure 1).

Blocking this pathway with JAK inhibitors results in the modulation of the gene expression of many inflammatory cytokines and enzymes. The CD4+ lymphocytes secrete IFN-γ (which activates macrophages) and other cytokines (including IL-2, IL-17, and monocyte-recruiting chemokines), while macrophages produce IL-6, IL-12, IL-18, IL-23, TNF-α, and T-lymphocyte chemokines [89,90]. Interactions between T lymphocytes and macrophages are probably responsible for the persistence of granulomatous inflammation [91]. Furthermore, Damsky et al. described the results of immunohistochemical studies performed on material taken from skin specimens from eleven NL patients, which showed an increased activation of STAT1 and STAT3 compared to specimens taken from healthy individuals [92]. The authors suggest that JAK-STAT signaling is activated at low levels in NL, so JAK inhibition may be an effective therapeutic option in this disease entity.

#### 6.3.2. Side Effects of JAK Inhibitors 

The most common side effect of JAK inhibitors is an increased risk of infections, especially of the upper respiratory tract and urinary tract [93,94]. Very rarely, opportunistic infections may also occur. Increased liver enzyme levels have also been reported in patients treated with JAK inhibitors, especially in combination with methotrexate and TNF-α inhibitors. Another side effect of JAK inhibitors is dyslipidemia, so it is recommended to examine the lipid profile during therapy. These drugs may also cause neutropenia and lymphopenia and increase the risk of thromboembolic events [93,95]. It has also been revealed that tofacitinib may increase the risk of malignancies (especially lymphomas and lung cancers) [96].

#### 6.3.3. Clinical Reports

Eight clinical case reports on the use of JAK inhibitors in patients with NL were found in the current literature. Four patients were successfully treated with tofacitinib, two with ruxolitinib, one with baricitinib, and one patient with abrocytinib.

#### 6.3.4. Ruxolitinib

The first case, described in 2018 by Lee et al., involved the administration of ruxolitinib (a JAK1 and JAK2 inhibitor) to a 71-year-old patient for the treatment of polycythemia vera (with a JAK2 gene mutation) [79]. In addition, the patient had type 2 diabetes mellitus, chronic renal failure, and multiple painful and ulcerated plaques located on the breast, abdomen, buttocks, and upper and lower limbs for 12 years. Histopathological examinations of a specimen from the lesion yielded a diagnosis of NL. The lesions were treated with topical GCS, hydroxychloroquine, and mycophenolate mofetil, without satisfactory improvement. Subsequently, the patient was treated with infliximab, but despite partial improvement, treatment was discontinued after 9 months due to severe congestive heart failure. After 3 months of oral ruxolitinib 10 mg twice daily, there was a significant clinical improvement (healing of all ulcers and resolution of pain and pruritus).

Nugent et al. described an interesting case of a 19-year-old NL patient who, despite only partial improvement after concomitant treatment with pentoxifylline and hydroxychloroquine (discontinued after three months due to a fainting episode) and 2% tofacitinib cream, showed very significant improvement after switching from tofacitinib cream to 1.5% ruxolitinib cream. The plaques were significantly reduced. The authors suggest that the reason for the different therapeutic responses to tofacitinib and ruxolitinib may be due to selective enzyme inhibition (tofacitinib—JAK1 and JAK3, while ruxolitinib—JAK1 and JAK2) [97].

#### 6.3.5. Tofacitinib

Damsky et al. described a case of a 25-year-old female patient with type 1 diabetes mellitus who had worsening, often ulcerative NL lesions on her lower legs for nine years. Topical and intralesional steroids and pentoxifylline failed to improve her condition. Ulceration healing was observed after 6 weeks of tofacitinib (JAK1 and JAK3 inhibitor) at 5 mg twice daily. After 9 months, the size of the plaques had not decreased, so triamcinolone 5 mg/mL intralesionally was added to the treatment, resulting in a reduction in inflammation and a reduction in the size of the plaques. The combination of tofacitinib with a glucocorticosteroid appeared to be more effective than monotherapy with a JAK inhibitor. The authors concluded that this may be due to the fact that glucocorticosteroids block the JAK-independent cytokine pathway (e.g., TNF-α), so the combination with tofacitinib, which blocks the JAK-dependent cytokine pathway, produces a synergistic effect [92].

McPhie et al. described a case of a 71-year-old man with diabetes mellitus and a 15-year history of progressive granulomatous skin disease with extensive ulcerations. The lesions were located on the neck, trunk, arms, and legs. Histopathological examinations confirmed NL. The patient had previously been treated unsuccessfully with topical and intralesional corticosteroids, cephalexin, hydroxychloroquine, acitretin, pentoxifylline, and ustekinumab. After 4 weeks of tofacitinib 5 mg twice a day, there was a marked improvement with a decrease in erythema and a reduction in lesions. The ulcers had completely healed, and there was a reduction in pain [81].

The successful use of tofacitinib in NL was also described by Janßen S et al. in a 48-year-old female with extensive, severely painful ulcers on the proximal surface of the lower legs in the course of NL, which had not responded to previous treatment (prednisolone, topical tacrolimus, adalimumab) [98]. The drug was administered at a dose of 5 mg twice daily, and a reduction in inflammation and gradual healing of the ulcers was quickly observed. In addition, a hair-containing punch graft transplantation of the skin was also performed to accelerate the healing process. After 5 months, the ulcer surface had epithelialized.

Erfurt-Berge et al. described a case of a 29-year-old woman with type 2 diabetes and a 10-year history of progressive NL. Although previous treatment (topical steroids, dapsone, compression stockings) had resulted in partial improvement, chronic ulcers on both lower legs remained resistant to treatment, and new lesions appeared on the dorsal surfaces of the feet. Twelve weeks after the initiation of tofacitinib at a dose of 5 mg twice daily, significant improvement was observed with ulcer healing and a reduction in erythema, inflammatory infiltration, and pain. At week 16, the dose was reduced to 5 mg once daily, and further improvement was observed [99].

#### 6.3.6. Baricitinib 

Barbet-Massin et al. described the use of baricitinib (JAK1/2 inhibitor) in a 64-year-old female patient with type 1 diabetes. The skin lesions had been treated for 2 years with topical corticosteroids and tacrolimus followed by oral methotrexate without success. During therapy, she developed rheumatoid arthritis (RA) with rheumatoid nodules on both hands, so baricitinib 4 mg daily was included. After 6 months, remission was achieved in not only in RA but also in NL [100].

#### 6.3.7. Abrocitinib 

Arnet L. et al. described the use of abrocitinib (a selective JAK1 inhibitor) in a 53-year-old woman with progressive necrobiosis lipoidica of the lower legs and forearm for 6 years. None of the previous therapies (topical steroids, calcineurin inhibitors, PUVA, hydroxychloroquine, fumaric acid esters) had been effective. The patient was started on abrocitinib at a dose of 200 mg daily. A gradual clinical improvement was observed. After 12 weeks, the abrocitinib dose was reduced to 100 mg daily. The authors expected a stronger effect of the proposed therapy, but the patient was very satisfied. It was speculated that abrocitinib may have a weaker effect on improving lesion healing than tofacitinib (a JAK1/3 inhibitor), which may suggest that other Janus kinases are also involved in NL pathogenesis [101].

### 6.4. Tapinarof

Tapinarof is a topical aryl hydrocarbon receptor agonist that both decreases TNF-α/IL-23/IL-17 levels and inhibits STAT-6 activation [102]. Although data are scarce, tapinarof appears to be a promising therapeutic option for NL due to its dual mechanism of action [103,104,105] (Figure 2).

Palomares S. et al. described the first case of the use of tapinarof in the treatment of NL in a 44-year-old female patient [106]. After one month of topical treatment with tapinarof in the form of a 1% cream applied to the lesions twice daily, a remission of the lesions was observed. To prevent recurrence, the patient applied tapinarof once daily only on weekends; however, new lesions developed and cleared within one week when the twice-daily application was resumed. The advantage of tapinarof over TNF-α inhibitors and JAK inhibitors is the topical application of the drug, which reduces the risk of side effects [106]. Although not seen in most patients, tapinarof may cause folliculitis, contact dermatitis, and headaches [107,108].

### 6.5. Other Methods

Other treatments are sometimes effective in the treatment of NL. Bacaranii et al. described the case of a 30-year-old female patient who underwent surgical resection of an NL-like lesion followed by a skin autograft with very good aesthetic results [109]. On the other hand, Fulgencio-Barbarin J. et al. suggested that the sequential punch grafting technique may be an effective adjunctive therapy for the treatment of refractory NL [110]. Hyperbaric oxygen therapy may also be beneficial [111]. In contrast, Abdat R. et al. used fractional microneedle radiofrequency effectively in two patients with NL. The therapeutic effect in these patients, however, may have resulted both from improved collagen and elastin deposition in the skin as a direct result of the treatment, as well as indirectly from the increased penetration of topical corticosteroid applied to the lesions after the treatments [112]. Motolese A. et al. demonstrated a very high efficacy of NL therapy with platelet-rich plasma applied to skin lesions in the form of a clot-like gel. Each of the 15 patients, despite previous resistance of the lesions to treatment, responded with complete clinical remission with the absence of side effects. The authors believe that this effect is mediated by growth factors and cytokines released by platelets that stimulate the healing of persistent NL wounds [113]. However, all of the therapeutic options mentioned here, although they have shown promising results, require further studies on larger groups of patients.

## 7. Conclusions

Necrobiosis lipoidica is a rare granulomatous disease that occurs more often in middle-aged women. Sharply demarcated, oval plaques with a raised erythematous border and yellowish-brown central atrophy are usually located bilaterally on the pretibial surfaces of the lower extremities. The diagnosis of NL is based on clinical presentation and histopathologic examinations. Dermoscopy may also be helpful. The differential diagnosis of other granulomatous diseases, which may have a similar clinical course, and squamous cell carcinomas should be considered. First-line treatment is usually topical or intralesional corticosteroids. Calcineurin inhibitors, especially tacrolimus, cyclosporine, antimalarial drugs, fumaric acid esters, and phototherapy, are also used, as well as a variety of physical modalities. In patients with refractory lesions, biologic drugs often prove effective. Recently, numerous attempts of newer therapeutic options in NL have been conducted. Highly promising trials of Janus kinase inhibitors or tapinarof, not registered for the treatment of this dermatosis yet, have helped worldwide. JAK inhibitors have been shown to be effective, especially in cases where biologic drugs were not. Therefore, these newest therapies, already approved and effectively used in other skin diseases, may become the best therapeutic path in NL, especially refractory to previous options, but the risk of side effects or contraindications should always be considered. Further multicenter studies on elucidating the etiopathogenesis of NL and exploring newer more effective treatment are needed to establish uniform therapeutic recommendations.

## Figures and Tables

**Figure 1 ijms-25-03482-f001:**
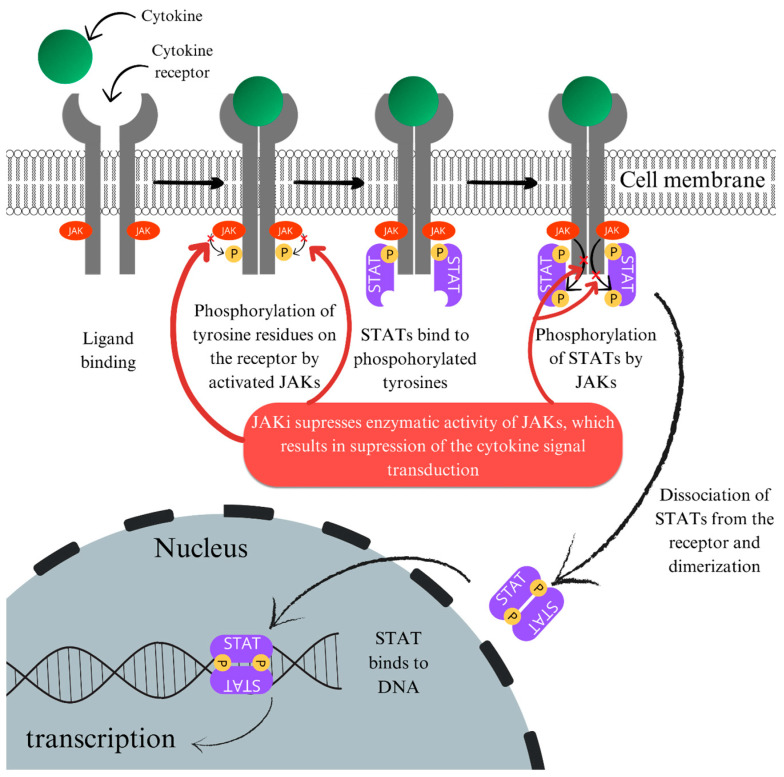
Mechanism of action of JAK inhibitors. (DNA—deoxyribonucleic acid). Based on Kisseleva et al., Roskoski et al. [87,88].

**Figure 2 ijms-25-03482-f002:**
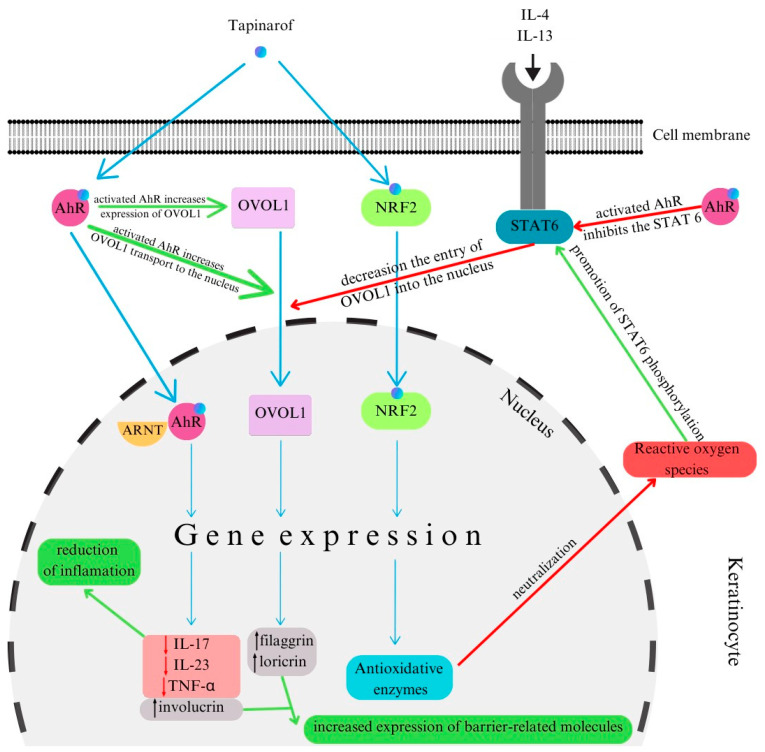
Mechanism of action of tapinarof. (Abbreviations: OVOL1—ovo-like transcriptional repressor 1; ARNT—aryl hydrocarbon receptor nuclear translocator; NRF2—nuclear factor erythroid 2-related factor 2). Based on Furue et al., Bissonnette et al. [103,104,105].

**Table 1 ijms-25-03482-t001:** Dermatoscopic observations—modified according to Shrestha et al. [52].

Trait	Observations	Possible Explanation
Vessel morphology	Initially, the linearly curved (comma-shaped) vessels become wavy and then tree-branched.	In early lesions, the comma-shaped vessels are due to vasodilatation in the papillary layer of the dermis as there are no epidermal changes. As the lesion progresses, the epidermis atrophies, resulting in the appearance of dilated vessels located in the deep layers of the dermis, which appear as linear–wavy and then tree-like branching.
Distribution of vessels	Vessels are evenly distributed.	-
Background	The vessels can be seen against a background of evenly spaced yellow areas without structure and whitish linear streaks.	Yellow areas without structure represent skin granulomas, while white linear streaks represent fibrosis.
Pigment network	In advanced lesions, brown pigmented networks may be visible.	This change is caused by the stimulation of melanocytes at the dermal–epidermal border. This phenomenon is nonspecific and common to many inflammatory skin lesions.

**Table 2 ijms-25-03482-t002:** Differentiating features between NL and other granulomatous diseases [58,59,60,61].

Disease	Clinical Description	Histological Description	Dermatoscopic Description
Necrobiosis lipoidica	Red papules or nodules, transform into round, oval plaques on the pretibial surface of the lower extremities; they enlarge peripherally, leaving a yellowish-brown atrophy in the central part with dilated small vessels.	Interstitial and palisade granulomas are visible. Lesions are layered and mixed with degenerated collagen. No increase in central mucin is observed. Plasma cells, multinucleated cells, and loss of elastic tissue are present.	Tree-shaped vessels of equal diameter without branching into finer capillaries. White linear streaks present. Lesions on a yellow-white background.
Granuloma annulare	Skin-colored papules, merging into ring-shaped foci. Various varieties possible.	Palisade granulomas with necrotic core of collagen and mucin can be seen. A lymphocytic infiltrate is present.	No visible vessels. Background in various shades of red, without structures on the periphery.
Sarcoidosis	Reddish-brown, round or ring-shaped plaque or nodular foci. Various variations possible.	In the specific type, there is a dermal infiltration of non-necrotic epithelioid granulomas. In the nonspecific type, the lesion is reactive. The epithelioid infiltration of histiocytes is present.	Vessels shorter and less branched than in NL. White reticulate streaks present. Orange globules in the background. Millia-like cysts also visible.
Necrobiotic xanthogranuloma	Hardened erythematous and yellowish plaques that may develop into scars, ulcers, and telangiectasias. The most common localization is on the face.	Widespread vitreous necrosis with foci of xanthogranulomatous infiltration in the reticular layer of the dermis into the subcutaneous fat. Cholesterol fissures with histiocytic infiltration with giant cells are visible.	Red-yellow area with irregular telangiectasias.
Localized scleroderma	Early lesions are erythematous. As the lesions progress, they become sclerotic and are surrounded by a “lilac” ring, and the center of the lesions is whitish or ivory in color.	In early lesions, the inflammatory margin shows an inflammatory infiltrate composed mainly of large numbers of lymphocytes and plasma cells. Sclerotic lesions show collagen fibers extending into the reticular layer of the dermis.	Whitish bundles of fibrosis that often cross linear branching vessels. Pigment network-like structures are also often visible.

**Table 3 ijms-25-03482-t003:** Clinical studies on the treatment of patients with NL with ustekinumab and secukinumab. Includes information on patient age and gender, presence of ulcers, treatment administered, response to treatment, and adverse events of ustekinumab and secukinumab.

References	Sex/Age	Ulceration	Earlier Treatment	Biological Treatment	Response to Biological Treatment	Adverse Events
Beatty et al. [80]	24/F	+	topical clobetasol propionate, intralesional triamcinolone, doxycycline	Ustekinumab 45 mg week 0, week 4 and every 12 weeks thereafter	after 4 weeks: re-granulation of the ulcerated plaque after 12 weeks: almost complete re-granulation	none
Pourang et al. [77]	29/F	+	topical, intralesional, and systemic steroids, topical tacrolimus, oral antibiotics, and antifungals, hydroxychloroquine, pentoxifylline	Ustekinumab 90 mg every 2 months	after a few months: lesions improved	cellulitis
				earlier: adalimumab	new plaques have appeared (treatment was discontinued due to adverse effects)	abdominal rash, urticarial plaques at the injection site, new plaques
Hassoun et al. [76]	42/F	+	pentoxifylline, cyclosporine, mycophenolate mofetil	Ustekinumab 45 mg every 9 weeks	significant reduction in pain and pruritus, absence of recurrent ulcerations	none
				earlier: infliximab	healing of ulcerations (treatment was discontinued due to adverse effects)	anaphylactoid reaction
				earlier: adalimumab	no response	none
				earlier: etanercept	etanercept was effective for 6 months, until loss of efficacy, recurrence of ulcerations	none
McPhie et al. [81]	71/M	+	topical and intralesional corticosteroids, cephalexin, hydroxychloroquine, acitretin, pentoxifylline	ustekinumab 90 mg every 8 weeks	no response	not mentioned
Gibson et al. [78]	45/F	+	none	secukinumab s.c. 300 mg weekly for 5 weeks then every 4 weeks (total: 24 weeks)	moderate improvement (about 50%)	toothache, dry socket reported; extraction of broken tooth, peritonsillar cellulitis
				earlier: adalimumab s.c 40 mg weekly (for several months)	no response	not mentioned
	30/F	-	topical corticosteroid	secukinumab s.c. 300 mg weekly for 5 weeks then every 4 weeks (total: 24 weeks)	very significant clearance (about 90%)	none
	38/F	+	topical corticosteroid, intralesional triamcinolone	secukinumab s.c. 300 mg weekly for 5 weeks then every 4 weeks (total: 24 weeks)	marked improvement (about 75%)	shortness of breath, possible anxiety attack reported at week 1
	56/F	+	phototherapy, clobetasol and other corticosteroids, hydroxychloroquine	secukinumab s.c. 300 mg weekly for 5 weeks then every 4 weeks (total: 16 weeks)	slight improvement (about 25%)	swollen feet at week 1, improvement at week 2

## Data Availability

Data are contained within the article.

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
