# Peer review of "A Breakthrough in the Treatment of Necrobiosis Lipoidica? Update on Treatment, Etiopathogenesis, Diagnosis, and Clinical Presentation"

_ijms, 2024, doi:10.3390/ijms25063482_

Round 1

Reviewer 1 Report

Comments and Suggestions for Authors

1. "11–87% of people with NL 45 are diagnosed with diabetes": why this big percentage? can you better explain the existing literature?

2. which is the prevalence of NL?

3. uncommon cases of pediatric NL have been reported --> Bonura C, et al. Necrobiosis Lipoidica Diabeticorum: A pediatric case report. Dermatoendocrinol. 2014 Jan 1;6(1):e27790. 

4. why BCC has been included among differential diagnoses of NL?

5. paragraph about conventional treatment should be improved

6. anti tnf alfa have been used in NL?

Author Response

Dear Reviewer,

Thank you for your valuable time, considering our manuscript for publication and reviewing our paper. We have improved the manuscript by answering all your questions. Please see the attachment:

''1. "11-87% of people with NL 45 are diagnosed with diabetes": why this big percentage? Can you better explain the existing literature?'' 

Thank you for your accurate comment. In the chapter on etiopathogenesis, we have highlighted precise data from a recent clinical case series and briefly discussed the reason for such a large spread of data.

“2. What is the prevalence of NL?”

We attempted to review the literature once again, to find the necessary information. Unfortunately, there is no summary of this issue as it is a rare diagnosis, and information on its prevalence is only available from clinical case series descriptions. In this paper, we have referred to all available data on epidemiology; these mainly relate to the co-occurrence of diabetes and other diseases.

“3. uncommon cases of pediatric NL have been reported --> Bonura C, et al. Necrobiosis Lipoidica Diabeticorum: A pediatric case report. Dermatoendocrinol. 2014 Jan 1;6(1):e27790.”

Thank you for bringing this important work to our attention. It has been added and discussed in the chapter describing the clinical picture.

“4. why has BCC been included among differential diagnoses of NL?”

Thank you for the comment. Due to the presence in BCC of tree-branched vessels also present in NL, as presented by Shrestha et al., we decided to include BCC in the differential diagnosis of NL. However, after reviewing other differences in these diseases, we decided not to include BCC in the differential diagnosis of NL and adjusted the manuscript.

“5. The paragraph about conventional treatment should be improved”

We have improved the entire section on conventional treatment, adding more information on, among other things, treatment with GCS, calcineurin inhibitors, and previously used biological drugs. We have added adequate citations.

“6. anti tnf alfa have been used in NL?”

Given the role of insulin resistance (added in the etiopathogenesis), we justified the rationale for using TNF-α inhibitors and briefly described them in the treatment section. In addition to the drugs mentioned in the manuscript, successful trials of pentoxifylline and thalidomide, which are also categorized as TNF-alpha inhibitors, have been described.

We hope that we answered all your questions satisfactorily,

Best regards;

Stefan Modzelewski and coauthors.

Reviewer 2 Report

Comments and Suggestions for Authors

The manuscript entitled „A breakthrough in the treatment of Necrobiosis Lipoidica? Update on treatment, etiopathogenesis, diagnosis, and clinical presentation.“ could be a valuable article which presents useful data on  necrobiosis lipoidica.

 In their review, the authors  present prominent features of this disease, including new therapeutic approaches (potentially effective in patients with refractory lesions), the presumed etiopathogenesis and diagnostic guidance for clinicians. As known, necrobiosis lipoidica  is a rare granulomatous disease of a not fully understood etiopathogenesis which is classically associated with insulin-dependent diabetes mellitus, which often fails to respond to conventional treatments. Thus, first  line medications are usually topical corticosteroids, but patients respond to them with varying degrees of success, while oother options include tacrolimus, phototherapy, cyclosporine, fumaric acid esters  and biologics (adalimumab, etanercept, infliximab).  In this review the authors concluded that Janus kinase  inhibitors and biologics such as ustekinumab and secukinumab can be used effectively in patients with recalcitrant necrobiosis lipoidica. Another promising treatment option is tapinarof (an aryl hydrocarbon receptor agonist), but there is a need for the studies on larger groups of patients - to evaluate the effectiveness of different therapeutic options and to define consistent treatment regimens for necrobiosis lipoidica.

This manuscript is valuable and may contribute to further work with patients with similar clinical pictures and situations. In addition, it includes nice figures. 

However, there are some suggestions.

 -Some parts of the text could be divided into specific paragraphs, because they are somewhat “mixed” and “jumped” in some places of the text, e.g. in the Introduction section. Thus, some paragraphs are too long and include lots of text, which is difficult to follow. So, I suggest dividing long paragraphs into smaller paragraphs according to a specific topic, with the aim to more clearly present the content.

-It is needed to delete the point (.) in the title (at its end)

- When talking about topical therapy, it is needed to mention intralesional corticosteroids because intralesional corticosteroid therapy is a common treatment for necrobiosis lipoidica. Also, when talking about corticosteroids, it is necessary to mention their adverse reactions when used on them for a long time (it is a common problem in the practice).

-The manuscript (Discussion section) involves the results of various authors who were mentioned and cited, but please shorten their texts where possible. 

- In the text after the subtitle “Treatment standards …” please add more references

-What is the basis for your figures (schemes)? Is it original? Do you use data from specific articles for its design? If yes, it is needed to mention them.

Author Response

Dear Reviewer,

Thank you for your valuable time and precise comments. We are delighted to know that our work has been found to be clinically useful. We have prepared responses to all your comments:

 ''-Some parts of the text could be divided into specific paragraphs, because they are somewhat "mixed" and "jumped" in some places of the text, e.g. in the Introduction section. Thus, some paragraphs are too long and include lots of text, which is difficult to follow. So, I suggest dividing long paragraphs into smaller paragraphs according to a specific topic, with the aim to more clearly present the content.''

We have followed your comment and removed repetitive information. In addition, due to the length of the paragraphs, we have condensed and rearranged them, especially the treatment sections.

''-It is needed to delete the point (.) in the title (at its end)''

Of course, the point has been deleted.

''- When talking about topical therapy, it is needed to mention intralesional corticosteroids because intralesional corticosteroid therapy is a common treatment for necrobiosis lipoidica. Also, when talking about corticosteroids, it is necessary to mention their adverse reactions when used on them for a long time (it is a common problem in the practice).''

Thank you for your advice. In the section on conventional treatment, we discussed the use of corticosteroids and the side effects of long-term treatment with them.

''The manuscript (Discussion section) involves the results of various authors who were mentioned and cited, but please shorten their texts where possible. ''

We shortened texts where it was possible. Please see the attachment.

''- In the text after the subtitle “Treatment standards …” please add more references''

We added more references and improved this section of our manuscript.

''-What is the basis for your figures (schemes)? Is it original? Do you use data from specific articles for its design? If yes, it is needed to mention them.''

Thank You for your accurate advice. We added references which was our inspiration for schemes.

We are grateful for your work and for drawing our attention to the important issues.

Best regards,

Stefan Modzelewski and coauthors.

Reviewer 3 Report

Comments and Suggestions for Authors

The review analysis the novel findings in necrobiosis lipoidica. The structure is very well written. the article is complex. The chapters are well written. The refferences are upt to date. Please discuss the impact of insulin resistance on this skin condition. please check:Bungau AF, Radu AF, Bungau SG, Vesa CM, Tit DM, Endres LM. Oxidative stress and metabolic syndrome in acne vulgaris: Pathogenetic connections and potential role of dietary supplements and phytochemicals. Biomed Pharmacother. 2023 Aug;164:115003. doi: 10.1016/j.biopha.2023.115003. Epub 2023 Jun 12. PMID: 37315434.

Comments on the Quality of English Language

The review analysis the novel findings in necrobiosis lipoidica. The structure is very well written. the article is complex. The chapters are well written. The refferences are upt to date. Please discuss the impact of insulin resistance on this skin condition. please check:Bungau AF, Radu AF, Bungau SG, Vesa CM, Tit DM, Endres LM. Oxidative stress and metabolic syndrome in acne vulgaris: Pathogenetic connections and potential role of dietary supplements and phytochemicals. Biomed Pharmacother. 2023 Aug;164:115003. doi: 10.1016/j.biopha.2023.115003. Epub 2023 Jun 12. PMID: 37315434.

Author Response

Dear Reviewer,

Thank you for your valuable time and your valuable commentary.

“Please discuss the impact of insulin resistance on this skin condition. please check:Bungau AF, Radu AF, Bungau SG, Vesa CM, Tit DM, Endres LM. Oxidative stress and metabolic syndrome in acne vulgaris: Pathogenetic connections and potential role of dietary supplements and phytochemicals. Biomed Pharmacother. 2023 Aug;164:115003. doi: 10.1016/j.biopha.2023.115003. Epub 2023 Jun 12. PMID: 37315434.”

Of course, we have referred to the mentioned article in the section on etiopathogenesis and evaluated the role of insulin resistance in necrobiosis lipoidica. Please see the attachment.

With best regards,

Stefan Modzelewski and co-authors.